# Masked Path Modeling for Vision-and-Language Navigation

**Zi-Yi Dou[†], Feng Gao[♯], Nanyun Peng[†]**
[†]University of California, Los Angeles   [♯]Amazon Alexa AI
{zdou,violetpeng}@cs.ucla.edu  fenggo@amazon.com

## Abstract

Vision-and-language navigation (VLN) agents are trained to navigate in real-world environments based on natural language instructions. A major challenge in VLN is the limited available training data, which hinders the models' ability to generalize effectively. Previous approaches have attempted to alleviate this issue by using external tools to generate pseudo-labeled data or integrating web-scaled image-text pairs during training. However, these methods often rely on automatically-generated or out-of-domain data, leading to challenges such as suboptimal data quality and domain mismatch. In this paper, we introduce a masked path modeling (MPM) objective. MPM pretrains an agent using self-collected data for subsequent navigation tasks, eliminating the need for external tools. Specifically, our method allows the agent to explore navigation environments and record the paths it traverses alongside the corresponding agent actions. Subsequently, we train the agent on this collected data to reconstruct the original action sequence when given a randomly masked subsequence of the original path. This approach enables the agent to accumulate a diverse and substantial dataset, facilitating the connection between visual observations of paths and the agent's actions, which is the foundation of the VLN task. Importantly, the collected data are in-domain, and the training process avoids synthetic data with uncertain quality, addressing previous issues. We conduct experiments on various VLN datasets and demonstrate the applications of MPM across different levels of instruction complexity. Our results exhibit significant improvements in success rates, with enhancements of 1.3%, 1.1%, and 1.2% on the val-unseen split of the Room-to-Room, Room-for-Room, and Room-across-Room datasets, respectively. Additionally, we underscore the adaptability of MPM as well as the potential for additional improvements when the agent is allowed to explore unseen environments prior to testing.[1]

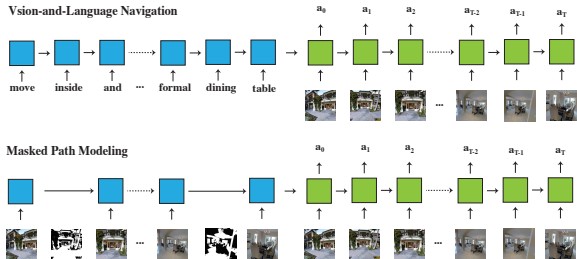

Figure 1: Vision-and-language navigation agents are trained to predict actions in a real-world environment given a natural language instruction. We incorporate our proposed masked path modeling objective into training, where the agent is trained to reconstruct the original action sequence when given a randomly masked subsequence of the visual inputs of the original path. The figure is adapted from Anderson et al. (2018b).

## 1 Introduction

A vision-and-language navigation (VLN) agent is trained to follow natural language instructions and navigate within an environment to achieve a specified goal. This task requires the agent to possess several sophisticated abilities, including understanding and grounding language phrases to visual objects, as well as planning and executing actions in a real-world setting.

The pretraining-then-finetuning paradigm (Peters et al., 2018; Devlin et al., 2019; Chen et al., 2021b; He et al., 2022) has proven to be effective in addressing these challenges in the field of VLN. By utilizing various supervision signals and proposing pretraining objectives, significant improvements have been demonstrated across VLN tasks. Prior works have explored the use of internet-scale image-text datasets to acquire grounded vision and language representations. Notable contributions in this area include the works of Majumdar et al. (2020) and Guhur et al. (2021), who leverage web-scraped image-caption corpora to learn

---

[1]Code is available at https://github.com/PlusLabNLP/mpm.

general vision-language representations and then finetune the models using in-domain VLN data to adapt the representations specifically for VLN tasks. Similarly, Shen et al. (2022) and Khandelwal et al. (2022) employ the CLIP vision encoder (Radford et al., 2021) pretrained with image-text contrastive loss and showcase the application of such models in various embodied tasks.

While the vision-language pretrained representations improve the alignment between vision and language modalities, it is important to note that there is a domain gap between the pretraining data and VLN data. In addition, these models are unaware of how to connect the learned *representations to actions* because they are not explicitly trained for *generating actions*, which is a critical skill for VLN tasks. To address the issue and integrate action generation into pretraining, prior works such as PREVALENT (Hao et al., 2020) HAMT (Chen et al., 2021a) use a single-step action prediction objective based on human-annotated and synthetic image-text-action triplets. However, the scalability of the pretraining objective is limited due to the scarcity of annotated instruction-action pairs. Specifically, they require training on pairs of natural language instruction and action sequences which are costly to obtain in a scalable way. There is also a line of work that utilizes synthetic data for training, either obtaining pseudo-instructions from sampled paths (Fried et al., 2018b; Tan et al., 2019; Hao et al., 2020; Wang et al., 2022b) or generating both the visual environments as well as the language instructions (Kamath et al., 2023; Wang et al., 2023). However, the automatically-generated data cannot be perfect and the noise during data generation can impact the model performance.

In this paper, we present an approach to pretrain VLN models with masked path modeling (MPM), which leverages in-domain path data collected by an agent for self-supervised learning. The proposed objective targets addressing the two major limitations of previous work:

- It collects scalable in-domain pretraining data without data synthesis. During the pretraining phase, the VLN agent explores the environment randomly and gathers navigation paths along with its actions, which are then used for pretraining the agent. Because the agent actively explores different environments, we can collect a rich amount of diverse paths. In addition, the MPM objective only requires the collected path and action information for training, eliminating the need for synthesizing additional signals.

- It explicitly focuses on conditional action generation. Concretely, as shown in Figure 1, to construct the MPM objective, we randomly mask certain viewpoints in the self-collected paths and train the agent to reconstruct the original paths based on the masked ones. MPM is similar to the VLN objective because the agent is trained to output a sequence of actions given specific instructions, with the distinction that the instructions are presented as masked paths rather than natural language instructions. Consequently, MPM effectively prepares the agent for VLN tasks that require conditioned action generation skills.

As a result, our pretraining objective is scalable and well-suited for addressing the VLN task.

We evaluate the proposed method on various VLN datasets with different types of instructions, including Room-to-Room (Anderson et al., 2018c), Room-for-Room (Jain et al., 2019), and Room-across-Room (Ku et al., 2020). Experimental results demonstrate that MPM can achieve significant improvements in both seen and unseen environments compared with strong baselines. For example, we achieve improvements of 1.32%, 1.05%, and 1.19% on the val-unseen split of the Room-to-Room, Room-for-Room, and Room-across-Room datasets, respectively. In addition, we demonstrate the MPM can be flexibly integrated into different types of models and achieve improvements. Furthermore, an analysis reveals the potential for additional improvements when the agent is allowed to explore unseen environments prior to testing.

## 2 Methods

In this section, we first introduce the basic settings and model architecture of vision-and-language navigation, then illustrate the details of each component of our proposed approach.

### 2.1 Background

**Training Data.** The training data $\mathcal{D}$ of VLN consists of parallel instruction-action pairs $\{(\mathbf{i}^k, \mathbf{a}^k)\}$ from different environments. However, it is hard to manually annotate a large amount of instruction-action data for VLN. Therefore, researchers have proposed various data augmentation strategies (Fried et al., 2018b; Tan et al.,

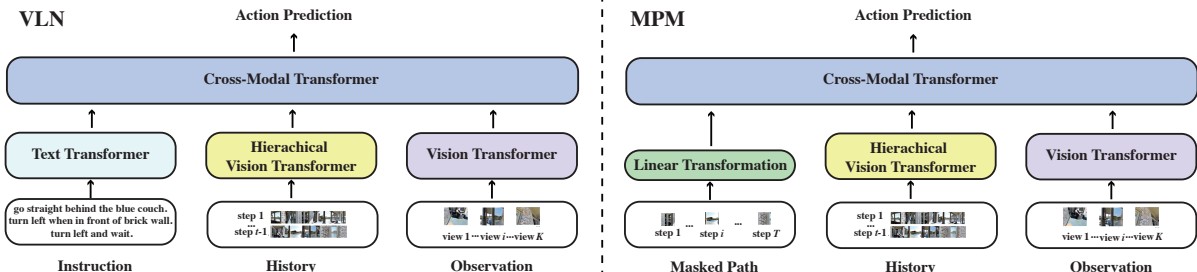

Figure 2: We follow Chen et al. (2021a) to design the base model architecture. In VLN, the model separately encodes the given instruction with a text encoder, its past history with a hierarchical visual encoder, and its current observations with another vision encoder; the encoded representations are then fed into a joint cross-modal transformer to predict the final action. In MPM, we directly feed a masked subpath to the cross-modal transformer instead of a language instruction and the model is trained to predict the original action given the masked subpath. All the parameters between VLN and MPM are shared.

2019; Li et al., 2022; Kamath et al., 2023) to provide additional supervision for VLN models. Int this work, we follow a common practice to use a speaker model (Fried et al., 2018b) to generate language instructions given randomly sampled paths and enrich the training data with the generated instruction-action pairs. Specifically, we incorporate the PREVALENT-generated data (Hao et al., 2020) into training.

**Base Settings.** Using the parallel instruction-action pairs $\mathcal{D}$, a VLN agent is trained to follow a given language instruction and generate a sequence of actions to reach the final destination in a photo-realistic environment. Formally, in a given environment $\mathbf{e}$, the navigation agent parameterized by $\theta$ learns to model the distribution $P(\mathbf{a}|\mathbf{i}, \mathbf{e}; \theta)$, where $\mathbf{i}$ and $\mathbf{a}$ denote instruction and action variables, respectively.

**Model Architecture.** In this paper, we employ a history-aware multimodal transformer architecture design following Chen et al. (2021a) as it achieves strong VLN performance across datasets, although it should be noted that our approach is compatible with most existing model architectures in VLN. Overall, as shown in Figure 2, at each action prediction step, we have a transformer text encoder to encode the given language instruction, a hierarchical vision transformer to encode all the past observations of the agent, and another vision transformer to encode the agent panoramic observation of the current step; then, the three types of representations will be concatenated together and fed into a cross-modal transformer for joint encoding, and the final pooled representation is used for action prediction.

**Text Features.** Following previous work (Chen et al., 2021a), we use pretrained text encoder to encode the language instructions. We use the standard BERT model (Devlin et al., 2019) to encode the English instructions for the R2R and R4R datasets (Anderson et al., 2018c; Jain et al., 2019) and the XLM-R model (Conneau et al., 2020) to encode the multilingual instructions for the RxR dataset (Ku et al., 2020).

**Vision Features.** At each step, the agent is given a panoramic observation of its current position in the environment, denoted as $\{v_i\}_{i=1}^K$. For each view in the panoramic observation, its vision feature is first extracted using a pretrained vision encoder. While many of the previous methods (Anderson et al., 2018b; Chen et al., 2021a) use vision encoders pretrained on ImageNet (Fei-Fei et al., 2009) for image classification, we find that CLIP vision encoder (Radford et al., 2021) achieves stronger performance, which is consistent with the findings of Shen et al. (2022). Therefore, we choose to use CLIP to first extract vision features and then the CLIP features are fed into the transformers for history and observation encoding.

For the current observations, in addition to the CLIP features, we also feed the model with the relative angle of each view $v_i$ in the panoramic observation, represented as $\text{REL}(v_i) = (sin\theta_i, cos\theta_i, sin\phi_i, cos\phi_i)$ where $\theta_i$ and $\phi_i$ are the relative heading and elevation angle to the agent's orientation, respectively. The combined representations $\{[\text{CLIP}(v_i); \text{REL}(v_i)]\}_{i=1}^K$ are then fed into a transformer to obtain $K$ processed representations.

**History Features.** The model also keeps track of its past observations with a hierarchical vision

transformer, where the panoramic observation at each step is first encoded by a single vector with a vision transformer, and all the panoramic representations are jointly encoded with another transformer along the temporal dimension. We refer to Chen et al. (2021a) for details.

**Cross-Modal Interactions.** The history features and the current observation features are concatenated as the vision modality, and a dual-stream cross-modal fusion architecture is used to encode both the vision and text modalities and allow for cross-modal information exchange. At each layer, we have a self-attention block for inter-modal interactions and a cross-attention block for vision-text interactions.

## 2.2 Masked Path Modeling

In this part, we illustrate the main idea of our proposed masked path modeling method. We go through the details of the active data collection, model architecture, and training strategies.

**General Framework.** Masked path modeling is inspired by the *masked data modeling* pretraining methods in the language and vision communities (Devlin et al., 2019; He et al., 2022), where the general idea is that the model is trained to reconstruct an original input (e.g., a sentence or an image) given parts of the input masked. In VLN, we propose to first ask an agent to perform a sequence of actions and collect a path consisting of several viewpoints $\mathbf{p} = \langle p_1, p_2, \cdots, p_n \rangle$, then we mask x%[2] of the viewpoints in this path and feed the observations along the masked path $\mathbf{p}_m = \langle p_{m_1}, p_{m_2}, \cdots, p_{m_k} \rangle$ to the agent and the agent is trained to perform the same sequence of actions as before to reconstruct the original path $\mathbf{p}$ given $\mathbf{p}_m$. Note that different from the common masked data modeling methods, *the input and output modalities are different in MPM*. Specifically, the model inputs are visual observations of the environment while the model outputs are the agent actions. This design can explicitly train the model to connect vision inputs and action outputs, which forms the foundation of the VLN task.

**Data Collection.** One of the major bottlenecks of VLN is the lack of training data and it is hard to collect large-scale in-domain data for VLN. In masked

path modeling, however, the agent can actively collect a great amount of data given an environment for training. During the data collection period, we ask the agent to randomly choose the next viewpoint with equal probabilities at each step. Also, we keep track of all the visited viewpoints, and the agent is not allowed to visit the same viewpoint twice. To control the length of the paths, we utilize pre-computed statistics from the training data regarding agent paths. We then randomly select paths, ensuring that each path length is sampled according to the distribution of path lengths observed in the training data. More sophisticated path collection techniques such as using techniques to encourage the diversity of sampled paths may also be used but here we leave it as future work. During path masking, we ensure that the last viewpoint is not masked so that the agent is always aware of the goal viewpoint.

**Model Architecture for MPM.** As in Figure 2, the main difference between masked path modeling and the VLN objective is that the input of masked path modeling is a sequence of visual observations instead of a natural language instruction. Therefore, we employ the same architecture as the original HAMT model except that we perform a linear transformation on the CLIP-encoded visual features so as to match the input and model dimensions, and then directly feed transformed features to the crossmodal transformer module. While collecting the visual features along a masked path, we do not use the panoramic view but only the view that the agent currently faces so as to make the pretraining task harder.[3] All the module parameters are shared between the masked path modeling and VLN objectives.

**Training Strategies.** We include our masked path modeling objective into the pretraining and finetuning stages of HAMT (Chen et al., 2021a). Concretely, during pretraining, the agent is jointly pretrained with masked path modeling and standard objectives including masked language modeling and instruction trajectory matching. We also include single-step action prediction and regression (SAP/SAR), and spatial relationship prediction (SPREL) objectives as in Chen et al. (2021a).[4]

---

[2] We set x=25 in this paper and perform sensitivity analysis in the experiment section.

[3] In the decoding side, the agent still receives panoramic views as in previous work (Fried et al., 2018b; Chen et al., 2021a).

[4] We do not use the masked region modeling objective in HAMT because it requires distilling the knowledge of an

| Model | Validation Seen | | | | Validation Unseen | | | | Test Unseen | | | |
|---|---|---|---|---|---|---|---|---|---|---|---|---|
| | TL | NE↓ | SR↑ | SPL↑ | TL | NE↓ | SR↑ | SPL↑ | TL | NE↓ | SR↑ | SPL↑ |
| HAMT (Chen et al., 2021a) | 11.15 | 2.51 | 76 | 72 | 11.46 | **2.29** | 66 | 61 | 12.27 | 3.93 | 65 | 60 |
| HAMT w/ Li and Bansal (2023) | - | - | - | - | - | - | 68 | 62 | - | - | 65 | 60 |
| HAMT+ | 11.11 | 2.65 | 75.02 | 71.75 | 11.93 | 3.34 | 67.05 | 61.69 | 12.70 | 3.57 | 67.19 | 61.94 |
| HAMT+ w/ MPM | 10.86 | **2.43** | **76.30** | **72.85** | 11.99 | 3.44 | **68.37** | **62.59** | 12.54 | **3.47** | 67.79 | **62.54** |

Table 1: Results on the Room-to-Room dataset (Anderson et al., 2018c). We incorporate MPM into a strong baseline (HAMT+) and achieve significant improvements across settings. The best scores are in **bold**.

| Model | Validation Seen | | | | | Validation Unseen | | | | |
|---|---|---|---|---|---|---|---|---|---|---|
| | NE↓ | SR↑ | CLS↑ | nDTW↑ | sDTW↑ | NE↓ | SR↑ | CLS↑ | nDTW↑ | sDTW↑ |
| HAMT (Chen et al., 2021a) | - | - | - | - | - | 6.09 | 44.6 | 57.7 | 50.3 | 31.8 |
| HAMT+ | 4.62 | 57.29 | 67.97 | 61.01 | 41.96 | 5.90 | 44.75 | 61.84 | 54.18 | 33.89 |
| HAMT+ w/ MPM | **4.29** | **59.13** | **70.50** | **64.88** | **48.28** | **5.65** | **46.88** | **62.76** | **55.23** | **35.50** |

Table 2: Results on the Room-for-Room dataset (Jain et al., 2019). MPM can also improve the model performance in this setting across all the evaluation metrics. The best scores are in **bold**.

The SAP and SAR objectives ask the model to predict the next action based on instruction, history from the ground-truth demonstration, and the current observation with imitation learning, where SAP formulates the task as a classification task while SAR trains the model to regress the action heading and elevation angles. The SPREL objective trains the model to predict the relative spatial position of two views in a panorama based on only visual features, angle features, or both types of features. We refer readers to Chen et al. (2021a) for more details.

Then, during finetuning, the model is jointly trained with both masked path modeling and the VLN objective with equal loss weights.[5] We combine the Asynchronous Advantage Actor-Critic (A3C) reinforcement learning objective (Mnih et al., 2016) and imitation learning objective for the VLN objective following previous work (Tan et al., 2019; Chen et al., 2021b), but only use the imitation learning objective for masked path modeling because it is stable and also it is non-trivial to design step-wise rewards in this setting.

## 3 Experiments

In this section, we present our experimental results with the proposed masked path modeling objective.

### 3.1 Settings

We go through the experimental settings in this part, including our used datasets, evaluation metrics, and implementation details.

#### 3.1.1 Datasets

We evaluate the models on different types of VLN datasets, including the Room-to-Room (R2R) (Anderson et al., 2018c), Room-for-Room (R4R) (Jain et al., 2019) and Room-across-Room (RxR) (Ku et al., 2020) datasets.

**R2R.** The R2R dataset is built based on Matterport3D (Chang et al., 2017) and has 7,189 paths, with each path paired with 3 different English instructions and the average length of all the paths is 29. R2R is split into training, validation, and test sets; the validation set consists of two splits: 1) *val-seen*, where all the paths are sampled from environments that are also seen in the training set, and 2) *val-unseen*, where paths are sampled from environments that do not appear in the training set so as to test the generalization ability of agents. The paths in the test set are from new environments unseen in the training and validation sets.

**R4R.** The R4R dataset is an algorithmically produced extension of R2R that concatenates two adjacent tail-to-head paths in R2R as well as their corresponding instructions to form a new instruction-path pair. With this extension, R4R has longer paths and instructions, and the paths are not always the shorted path from the starting point to the goal, making the dataset less biased than R2R.

image classification model while our vision encoder is CLIP. Also, the CLIP vision encoder is frozen during pretraining to save computation time.

[5]We did not see significant performance differences when tuning the loss weights in our preliminary studies.

| Model | Validation Seen | | | | Validation Unseen | | | | Test Unseen | | | |
|---|---|---|---|---|---|---|---|---|---|---|---|---|
| | SR↑ | SPL↑ | nDTW↑ | sDTW↑ | SR↑ | SPL↑ | nDTW↑ | sDTW↑ | SR↑ | SPL↑ | nDTW↑ | sDTW↑ |
| HAMT (Chen et al., 2021a) | 59.4 | 58.9 | 65.3 | 50.9 | 56.5 | 56.0 | 63.1 | 48.3 | 53.12 | 46.62 | 59.94 | 45.19 |
| HAMT+ | 63.93 | 59.93 | 68.59 | 55.47 | 62.00 | 58.05 | 67.52 | 53.87 | - | - | - | - |
| HAMT+ w/ MPM | **67.73** | **63.89** | **71.02** | **58.86** | **63.51** | **59.24** | 67.71 | **54.53** | 60.00 | 52.52 | 63.97 | 51.13 |

Table 3: Results on the Room-across-Room dataset (Ku et al., 2020). The best scores are in **bold**.

| Model | R2R Validation Unseen | | | | R4R Validation Unseen | | | |
|---|---|---|---|---|---|---|---|---|
| | TL | NE↓ | SR↑ | SPL↑ | NE↓ | SR↑ | CLS↑ | nDTW↑ | sDTW↑ |
| HAMT+ w/ MPM | 11.99 | 3.44 | 68.37 | 62.59 | 5.65 | 46.88 | 62.76 | 55.23 | 35.50 |
| HAMT+ w/ MPM-Prexplore | 11.37 | **3.33** | **69.60** | **64.69** | **5.13** | **51.34** | **63.72** | **57.95** | **39.06** |

Table 4: Pre-exploring the test environments with MPM can further improve the model performance.

**RxR.** The RxR dataset follows the same environment division as that in the R2R dataset. Different from R2R, RxR is a larger dataset that has 16,522 paths in total. In addition, the instructions are multilingual and in three languages, including English, Hindi, and Telugu. The lengths of the instructions in RxR are also much larger than that in R2R (average length: 78 vs. 29).

### 3.1.2 Evaluation Metrics

We adopt the standard evaluation metrics in VLN (Anderson et al., 2018a) to evaluate models. Specifically, we evaluate models with 1) trajectory lengths (*TL*): the length of the agent path measured in meters; 2) navigation error (*NE*): the average distance between the final position of agents and the goal position measured in meters; 3) success rate (*SR*): the proportion of agents whose final position is within three meters of the target; 4) success rate weighted by normalized inverse path length (*SPL*): success rate normalized by the ratio between the length of the shortest path and the predicted path.

Because the above metrics are heavily biased towards whether or not the agent can reach the goal position while ignoring the specific path the agents take, Jain et al. (2019) propose the coverage weighted by length score (*CLS*) metric that measures the path fidelity between the predicted path and target path for the R4R dataset. Similarly, Ku et al. (2020) propose normalized dynamic time warping (*nDTW*) and success rate weighted by dynamic time warping (*sDTW*) (Magalhães et al., 2019) for RxR.

### 3.1.3 Implementation Details

**Model Architecture.** We build our models upon the HAMT model (Chen et al., 2021a) and follow all of its parameter settings except that we

use CLIP-ViT (Radford et al., 2021) pretrained vision encoder and it is not finetuned during training. Specifically, our model consists of a 9-layer text transformer, a 2-layer panoramic transformer for encoding history information, and a 4-layer transformer for cross-modal encoding. In each panoramic observation, there are $K = 36$ views of images and we use CLIP-ViT-L-336/14 to encode the input images. We denote the HAMT baseline with CLIP-ViT-L-336/14 as **HAMT+**.

**Pretraining.** During pretraining, we randomly select proxy tasks including masked path modeling for each mini-batch with a predefined ratio as in Chen et al. (2021a). Different from Chen et al. (2021a), the CLIP-ViT is frozen instead of finetuned during both pretraining and finetuning in order to save computational costs. We train the model for 200k iterations with the AdamW optimizer (Loshchilov and Hutter, 2018) and the learning rate is set to 5e-5 and the batch size is set to 64. It take around 1 day to finish training on 4 NVIDIA Tesla V100 GPUs.

**Finetuning.** During finetuning, the model is jointly finetuned with the IL+RL and masked path modeling objectives with equal weights. The model is fine-tuned for 300k iterations with a learning rate of 1e-5 and batch size of 8 on a single V100 GPU, taking around 2.5 days to finish.[6] The best model is selected according to performance on the val unseen split. We use the same augmented data as Hong et al. (2021) following previous work for the R2R dataset, while no augmented data is used for other datasets. Greedy search is applied in inference following the single-run setting. In

---

[6]Following the hyper-parameters in `https://github.com/cshizhe/VLN-HAMT/blob/main/finetune_src/scripts/run_r2r.sh`.

| Path Design | R2R Validation Unseen | | | | R4R Validation Unseen | | | | |
|---|---|---|---|---|---|---|---|---|---|
| | TL | NE↓ | SR↑ | SPL↑ | NE↓ | SR↑ | CLS↑ | nDTW↑ | sDTW↑ |
| MPM w/ R2R Paths | 11.99 | 3.44 | **68.37** | **62.59** | 5.74 | 45.39 | 61.42 | 54.42 | 33.92 |
| MPM w/ R4R Paths | 11.97 | **3.38** | 68.20 | 62.30 | **5.65** | **46.88** | **62.76** | **55.23** | **35.50** |

Table 5: MPM performs the best when its collected paths resemble the paths of test environments. Here we only control the lengths of the paths to be similar to the paths of either R2R or R4R.

| Model | MPM | | R2R Validation Unseen | | | | R4R Validation Unseen | | | |
|---|---|---|---|---|---|---|---|---|---|---|
| | PT | FT | TL | NE↓ | SR↑ | SPL↑ | NE↓ | SR↑ | CLS↑ | nDTW↑ | sDTW↑ |
| HAMT+ | ✗ | ✗ | 11.93 | **3.34** | 67.05 | 61.69 | 5.90 | 44.75 | 61.84 | 54.18 | 33.89 |
| HAMT+ | ✗ | ✓ | 11.84 | 3.40 | 67.65 | 61.74 | 5.83 | 46.35 | **63.66** | **56.73** | **35.87** |
| HAMT+ | ✓ | ✓ | 11.99 | 3.44 | **68.37** | **62.59** | **5.65** | **46.88** | 62.76 | 55.23 | 35.50 |

Table 6: Including MPM during both pretraining (PT) and finetuning (FT) can generally lead to the best performance.

both pretraining and finetuning, the agent samples a batch of paths for MPM from the available environments in Matterport3D and we do not allow the agents to explore test environments.

## 3.2 Main Results

The main results of the baselines and our model are listed in Table 1, 2, and 3. We report both the numbers in the HAMT paper (Chen et al., 2021a) and our reproduced performance.

First, it should be noted that because we use the strong CLIP vision encoder, our reproduced HAMT+ baseline can achieve better performance than the original HAMT paper across settings, even surpassing the state-of-the-art end-to-end trained model (Li and Bansal, 2023). Especially, on the RxR datasets, our HAMT+ outperforms HAMT by 4.53% and 4.57 sDTW on the validation seen split and 5.5% success rate and 4.42 sDTW on the validation unseen split.

Built upon a strong baseline, our model can still outperform it across settings. Notably, the performance gains are pronounced when measured with path fidelity metrics (i.e., CLS, nDTW, sDTW) on the long-horizon VLN datasets R4R and RxR in the seen environments, indicating that the masked path modeling objective can encourage the models to faithfully follow the natural language instructions and complete the paths accordingly. This is intuitive as the VLN training objectives can optimize the models towards taking the shortest path to reach the final goal, whereas during masked path modeling, the model is trained to reconstruct the original paths, thus the model can follow the instructions more faithfully.

In unseen environments, our model achieves 1.32%, 2.13%, and 1.51% improvements over the

baseline in success rates on the R2R, R4R, and RxR datasets respectively on the validation set, demonstrating the effectiveness of our approach. We attribute these improvements to the fact that our objective allows the model to be trained on a variety of paths and can thus improve the generalization ability of the model in unseen environments.

## 3.3 Analysis

In this part, we perform several analyses to gain insights into MPM. We leave more analysis results in Appendix.

**Exploring Unseen Environments.** Because we allow the agents to autonomously acquire data and learn without the need for annotated data, we can train the agents in a scalable way. We hypothesize that when trained with masked path modeling, the agent can be familiarized with the explored environments and thus improve its navigation performance, even though it is not explicitly trained with the VLN objective. To verify this, we train the model with masked path modeling on unseen environments in the validation sets with the same hyper-parameters as before and test its VLN performance. As shown in Table 4, performing masked path modeling on unseen environments can significantly improve the model navigation performance, demonstrating the potential of using the objective in a large-scale setting. Especially, exploring unseen environments can bring 4.46% and 3.56 improvements in SR and sDTW on the R4R validation unseen set respectively.

**Path Design.** When collecting the paths, we make the lengths of the sampled paths follow the distribution of that in the training data so that the paths are similar to the navigation paths. As shown

| Model | R2R Validation Unseen | | | | REVERIE Validation Unseen | | | | | |
|---|---|---|---|---|---|---|---|---|---|---|
| | TL | NE↓ | SR↑ | SPL↑ | NE↓ | OSR↑ | SR↑ | CLS↑ | RGS↑ | RGSPL↑ |
| DUET (Chen et al., 2022) | 13.94 | 3.31 | 72 | **62** | 22.11 | 51.07 | 46.98 | 33.73 | 32.15 | 23.03 |
| DUET w/ MPM | 13.37 | **3.05** | **72.84** | **62.09** | 21.35 | **52.51** | **49.11** | **35.64** | **32.52** | **23.48** |

Table 7: MPM can be flexibly applied to various models and datasets.

in Table 5, if we switch the path designs between R2R and R4R, the performance gains will drop marginally, indicating that making the paths between MPM and VLN similar can best utilize the MPM objective for VLN. We leave more sophisticated path designs as future work.

**MPM during Pretraining.** During the pretraining stage, we follow HAMT to train the models with masked language modeling, instruction trajectory matching, single-step action prediction and regression, and spatial relationship prediction tasks. We also choose to include the masked path modeling objective during pretraining so as to mitigate the difference between pretraining and finetuning. As shown in Table 6, we can see that including masked path modeling is important as it can well prepare the models for the finetuning stage, although only doing masked path modeling during finetuning can also bring marginal improvements. Notably, not including MPM during pretraining seems to achieve comparable or even better performance than including it on R4R. One possible explanation is that during pretraining the path lengths are similar to those of R2R, thus the pretrained agent may be more suitable for R2R than R4R.

**Applications in Other Models and Datasets.** Previously, we mainly experiment with the end-to-end HAMT model on navigation tasks with fine-grained language instructions. In this part, we implement MPM upon the strong DUET model (Chen et al., 2022). As shown in Table 7, MPM can still improve the model performance with this map-based model on both R2R and REVERIE, demonstrating the flexibilty of MPM. It is worth noting that REVERIE is a remote object grounding task while MPM mainly concerns with navigating to a goal viewpoint or image, thus the object grounding performance is not improved signficantly, and we leave this as a future work.

## 4 Related Work

We overview three lines of related research, including vision-and-language navigation in general,

vision-and-language pretraining with a focus on its applications in VLN, as well as pretraining for control and embodied learning.

**Vision-and-Language Navigation.** Building vision-and-language navigation models has received increasing attention in recent years (Anderson et al., 2018b; Fried et al., 2018a; Wang et al., 2018; Li et al., 2019b; Zhu et al., 2020b; Kurita and Cho, 2021) and various benchmarks have been proposed to evaluate the ability of embodied agents to follow instructions and accomplish specified tasks (Kolve et al., 2017; Anderson et al., 2018a; Savva et al., 2019; Anderson et al., 2018c; Chen et al., 2019; Ku et al., 2020; Shridhar et al., 2020; Padmakumar et al., 2022). In this line of research, representative works include Fried et al. (2018b) who propose panoramic action space and use a speaker follow to synthesize additional training data. In addition, Tan et al. (2019) propose to mix imitation learning and A3C (Mnih et al., 2016) and increase the diversity of the synthesized data by adding noise into the environments during data generation. To utilize additional training signals, Ma et al. (2019) propose the self-monitoring agent that improves vision-language alignment with a co-grounding module and progress monitor; Zhu et al. (2020a) propose four self-supervised auxiliary tasks that are beneficial for the task of VLN. There are also works on designing better architectures (Chen et al., 2021a, 2022) for VLN. In terms of data augmentation, in addition to methods (Fried et al., 2018b; Tan et al., 2019; Hao et al., 2020; Dou and Peng, 2022) that synthetic language instructions are generated from randomly sampled paths, there are also works on synthesizing both the environments and the instructions (Wang et al., 2022b; Kamath et al., 2023). However, most of these works require external models to generate imperfect data, which can potentially harm the performance of VLN models.

**Vision-and-Language Pretraining.** Pretraining models on large web corpora have proven to be

highly effective in natural language processing (Peters et al., 2018; Devlin et al., 2019; Liu et al., 2019; Brown et al., 2020), and similar techniques have been applied in computer vision (Chen et al., 2020a, 2021b; He et al., 2022; Bao et al., 2022) and vision-language communities (Li et al., 2019a; Chen et al., 2020b; Radford et al., 2021; Kim et al., 2021; Li et al., 2021; Dou et al., 2022). In the field of VLN, researchers have tried to use unimodally or multimodally pretrained language or vision representations (Anderson et al., 2018b; Li et al., 2019b). Notably, the CLIP vision encoder (Radford et al., 2021) pretrained on large image-caption data has proven to be generally effective for vision-and-language embodied tasks (Khandelwal et al., 2022; Shen et al., 2022). To jointly learn transferrable vision and language representations for VLN, Majumdar et al. (2020) and Guhur et al. (2021) propose to first pretrain models on large image-caption data such as Conceptual Captions (Sharma et al., 2018) and then adapt the representations for VLN tasks by finetuning the model on in-domain VLN data. While the pretrained representations can be useful, the pretraining process does not explicitly connect the learned representations to output actions. To further integrate action generation into VLN pretraining, researchers have attempted to directly use VLN data for pretraining. For example, PREVALENT (Hao et al., 2020) is pretrained on image-text-action triplets with a single-step action prediction objective and masked instruction modeling objective; Chen et al. (2021a) further propose a single-step regression and spatial relationship prediction objective that introduces more supervisions. However, the pretraining data is limited by the size of VLN data and thus it is difficult to apply their approaches in large-scale settings; Li and Bansal (2023) propose three proxy tasks, one of which is to reconstruct the trajectory semantic information given masked inputs, but they fail to connect vision and language modalities and train on self-collected data.

**Pretraining for Control and Embodied Learning.** The general idea of pretraining on large-scale data has been adopted in embodied tasks, including data collection (Burda et al., 2019), representation learning (Yang and Nachum, 2021), and world model learning (Seo et al., 2023). For example, Pathak et al. (2017) present a curiosity-driven self-supervised data collection approach that encourages agents to explore unfamiliarized states; Nair et al. (2022) pretrain representations on human ego-centric video data and adapt the representations on robotic tasks. In terms of using self-collected data for training, Chaplot et al. (2021) propose to build 3D semantic maps and use them to improve action and perception skills. Similar to our work, Wu et al. (2023) propose to pretrain models by reconstructing a full trajectory or given a random subset of the same trajectory, and the pretrained models can be used for different downstream purposes like inverse dynamics, forward dynamics, imitation learning, offline RL, and representation learning. Different from these works, our method explicitly optimizes models for conditioned action generation and the agent can self-collect a rich amount of data for pretraining.

## 5    Conclusion

In this paper, we propose masked path modeling, a pretraining objective designed for vision-and-language navigation. The objective can utilize scalable data explored from different environments to improve the agent's conditioned action generation ability. We incorporate the objective into a strong baseline and demonstrate improvements across different settings. An analysis further reveals the potential for scaling the objective to large-scale settings. Future directions include designing better exploration strategies as well as investigating applications in more fields.

## Limitations

MPM is designed to connect vision and action modalities, while focusing less on the language modality. In addition, we constrain the agent to collect paths from the available environments in Matterport3D, which can limit the diversity of the paths to some degree. We also focus solely on VLN settings and do not investigate its applications in other embodied instruction following tasks.

## Acknowledgment

We thank anonymous reviewers for their insightful feedback. We also thank Cheng-Fu Yang, Sidi Lu and other members from the UCLA NLP group for their helpful feedback and discussions. Zi-Yi is supported by the Amazon Fellowship and DARPA Machine Common Sense (MCS) program under Cooperative Agreement N66001-19-2-4032 and NIH R01HL152270.

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

# A  Additional Analysis Results

In this section, we present more analysis results to provide further insights into our proposed method.

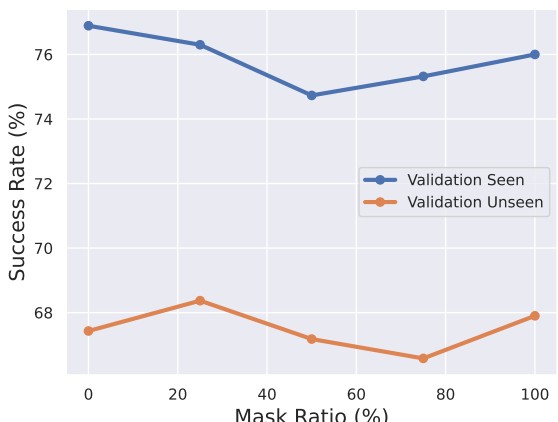

Figure 3: Effect of the mask ratio of the MPM objective. The model is generally robust to this hyper-parameter, with 25% achieving the best performance on unseen environments.

**Impact of the Mask Ratio.**  The mask ratio is a hyperparameter in masked path modeling and we investigate the optimal ratio in this paragraph. As shown in Figure 3, the objective is generally robust to the mask ratio, with 25% leading to the best performance and <=50% bringing improvements on the baseline. Therefore, we choose to randomly mask 25% of the viewpoints along a path in this paper. It is worth noting that because we always keep the goal viewpoint as mentioned in the method section, masking 100% of the paths is equivalent to an image-goal navigation task. Because the model performance can be improved even with 100% viewpoints removed, the improvements of MPM can be partially attribute to multi-task learning, which has proven to be effective in Wang et al. (2022a).

**Performance on Instructions of Different Lengths.**  We evaluate the performance of both the baseline model and our proposed approach on instructions of varying lengths on the R2R validation set. Figure 4 illustrates the results, indicating that the benefits of employing MPM are particularly significant when instructions are lengthy. We attribute this observation to the fact that the sampled paths occasionally involve intricate navigation patterns. Consequently, the integration of MPM enables the model to effectively learn how to navigate and follow complex instructions. Similarly,

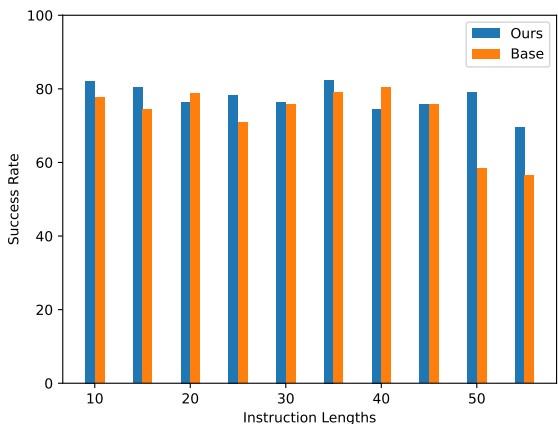

Figure 4: Performance of models on instructions of different lengths on the R2R validation set. Our model can significantly improve the model performance, especially on long instructions.

our method can achieve significant improvements when the instructions are relatively short possibly because the sampled paths are sometimes simple.

**Qualitative Examples.**    We also sample some examples from the R2R validation set with both the baseline model and our model. As shown in Figure 5 and 6, our method can encourage the model to closely follow the language instructions even if they are rather lengthy or concise. On the other hand, the baseline model fails to do so: it can generate incorrect paths when the instructions are short and false and unnecessarily complicated paths when the instructions are long.

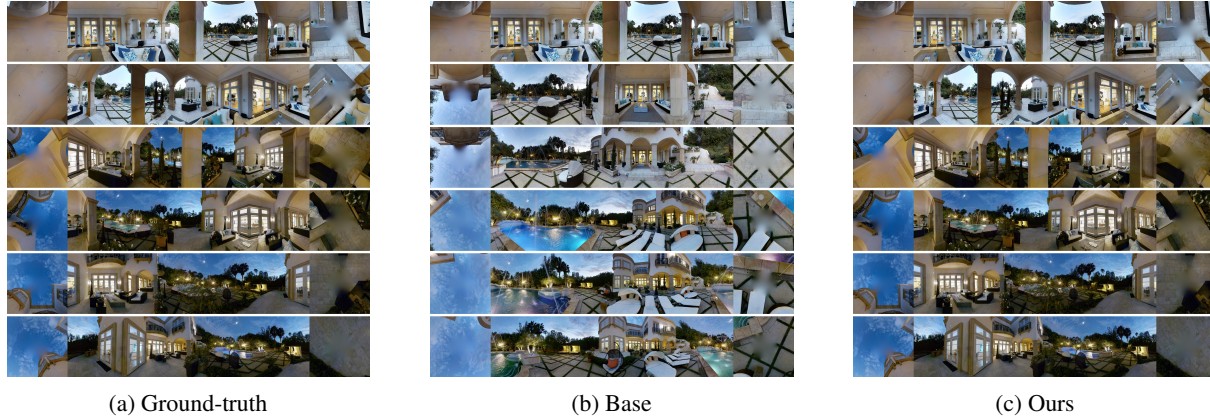

(a) Ground-truth        (b) Base        (c) Ours

Figure 5: Qualitative results for the instruction "Walk across patio, stop at hanging basket chair."

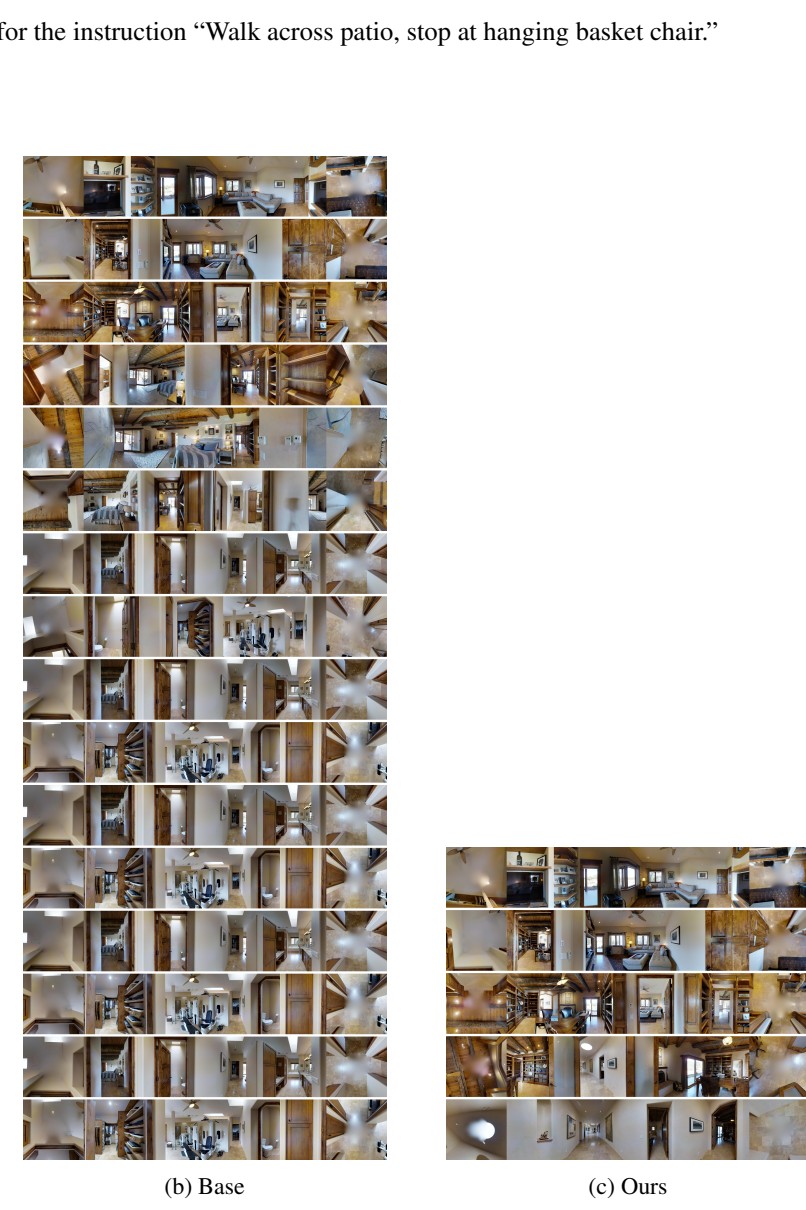

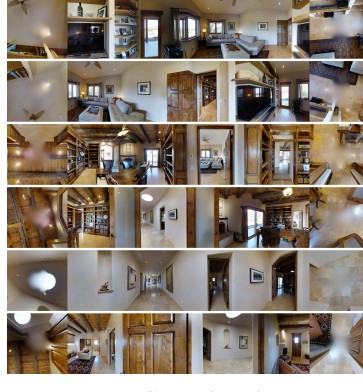

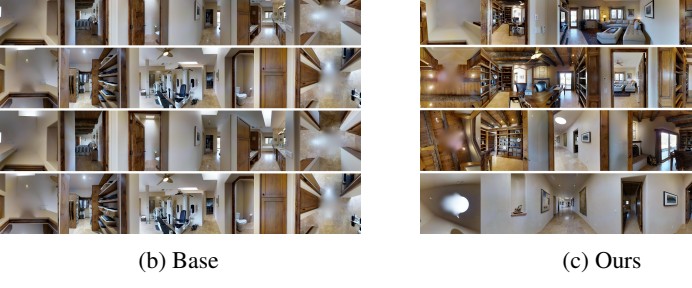

(a) Ground-truth        (b) Base        (c) Ours

Figure 6: Qualitative results for the instruction "Turn and walk towards the open brown wooden door that leads to an office with a large desk. Exit the room through the door. Walk around the left side of the table and go through the double open doors that leads to a hallway. Walk out into the hallway until you reach the first door on the right. Turn tight and take two steps into the room, stopping in the doorway to the room next to the carpet."