# OpenReview forum: "Masked Path Modeling for Vision-and-Language Navigation"
_EMNLP/2023/Conference — EMNLP 2023 Findings_

### Official Review · Reviewer_otNW · 2023-08-03

**Soundness:** 3

**Excitement:**

3: Ambivalent: It has merits (e.g., it reports state-of-the-art results, the idea is nice), but there are key weaknesses (e.g., it describes incremental work), and it can significantly benefit from another round of revision. However, I won't object to accepting it if my co-reviewers champion it.

**Paper Topic And Main Contributions:**

This paper proposes Masked Path Modeling to improve VLN performance. Specifically, this paper uses masked paths to replace the instructions and optimize the single action prediction training objective. Empirically, on R2R, R4R, and RxR datasets, the proposed approach improves the baseline method by around 1-2% in success rate.

**Questions For The Authors:**

N/A

**Reasons To Accept:**

1. Good performance on multiple VLN datasets.
2. The paper is well-written and easy to follow.
3. Reasonable approach to improve the performance.

**Reasons To Reject:**

1. The data collection pipeline is not new. In PREVALENT, they traverse through all the possible paths (with length ranging from 3-6) in the seen environments. Furthermore, PREVALENT generates synthetic instructions for all those unannotated paths. Given the data collection process mentioned in the paper, I don't think any new paths are actually created, thus I'm not sure why MPM helps reduce the data scarcity problem in VLN. Besides, no stats about the collected paths are given in the paper.
2. Simply adding a pre-training/fine-tuning objective is not novel enough. Similar ideas that optimize the self-supervised image-level loss are explored in "Improving Vision-and-Language Navigation by Generating Future-View Image Semantics", where they propose Masked Trajectory Modeling (MTM) that masks out views in a trajectory.
3. HAMT is a sota approach in 2021. To demonstrate the effectiveness of the proposed approach (i.e., could generalize to different model architecture/training strategy), at least MPM should be applied to "Think Global, Act Local: Dual-Scale Graph Transformer for Vision-and-Language Navigation" (CVPR 2022).

**Reproducibility:**

4: Could mostly reproduce the results, but there may be some variation because of sample variance or minor variations in their interpretation of the protocol or method.

**Reviewer Confidence:**

5: Positive that my evaluation is correct. I read the paper very carefully and I am very familiar with related work.

---

> ### Author Rebuttal · Authors · 2023-08-29
>
> Thank you for your comments! We appreciate that you acknowledge the good performance of our model and that our paper is well-written. Below we address your concerns:
>
> >Difference between PREVALENT and ours
>
> Thank you for pointing out the previous work! We would like to note that **PREVALENT requires an external model to generate pseudo language instructions** which can sometimes be of low quality. On the other hand, our method only requires the model to predict a self-collected path from the environment given some of its viewpoints masked. Therefore, the agent can collect a large number of paths from the environment without using an external model, ensuring the quality of the data. At each training step, we will sample a batch of paths for MPM and the number of the total sampled paths is 8*300K=2.4M. In addition, our pipeline is more flexible as we can collect paths in varying lengths (3.3 Path Design) whereas an external model can perform badly on out-of-domain paths (e.g. too long or short instructions).
>
> In addition, **our work is complementary to PREVALENT** because following previous work, we also use the PREVALENT-generated data (line 414-415) and we can still see performance improvements across settings, demonstrating that our work is indeed complementary to PREVALENT and both of them can improve the model performance.
>
> We will include this discussion in the revised version.
>
> >Masked Trajectory Modeling (MTM) in Li & Bansal (2023)
>
> Thank you for mentioning the work! We will cite this paper and compare with it in the revised version, and we would like to note that:
>
> - We respectfully disagree with summarizing our work as “simply adding a pretraining/finetuning objective”. In the introduction part, we well-motivated our work as pointed out by Reviewer rViw and emphasized that previous works relied on limited VLN data whereas our work can utilize large-scale self-collected data on conditional action generation (line 85-111), which none of the previous works have proposed before.
>
> - Similar to PREVALENT that also employs an MTM-like objective, their agent is **conditioned on both the instruction and paths** while our MPM objective only requires the model to be conditioned on **self-collected paths.** This design leads to significant differences: their model can only be trained on parallel instruction-path data which is limited, whereas our MPM allows the model to utilize large-scale self-collected data. We have this discussion in the Intro (line 74-80) and related work sections (line 577-586) and we will expand the paragraphs.
>
> - **Our model can outperform Li & Bansal (2023).** Following your suggestions, we perform additional experiments with DUET as the base model and below is the model performance on the R2R val-unseen set with minimal parameter tuning. We can see that our model outperforms Li & Bansal (2023). Considering that Li & Bansal (2023) have multiple pretraining objectives included (i.e., MTM, MPM, APIG), the performance gains demonstrated the advantages of MPM over previous work
>
> | **Model** | **TL** |  **NE** |  **SR** | **SPL**  |
> | :--------: | :--------: | :--------: | :--------: | :--------: |
> | DUET | 13.94 | 3.31 | 72 | 60 |
> | DUET + Li & Bansal (2023) | - | - | 72| 62 |
> | DUET+MPM | 13.61 | **2.90** | **72.80** | **63.64** |
>
>
> - We will cite and discuss the paper in detail but per the *ACL policies (https://www.aclweb.org/adminwiki/index.php?title=ACL_Policies_for_Submission,_Review_and_Citation), **papers appearing less than 3 months before the submission deadline should be considered contemporaneous to the submission**, and Li & Bansal (2023) is posted ~2 months before the submission date.
>
> > Stronger Baselines
>
> Thank you for your suggestions! We would like to note that:
>
> - Our work is based on an improved version of HAMT (referred to as HAMT+ in the paper, line 428-435). As reported in the paper, **compared with SoTA models without using the map information (e.g. Li & Bansal (2023)), HAMT+ can achieve comparable or even better performance.** Therefore, **our baseline is indeed strong** and improvements over this strong baseline should be convincing.
>
> | **Model** |  **Val Unseen SR** | **Val Unseen SPL**  | **Test Unseen SR** | **Test Unseen SPL**  |
> | :--------: | :--------: | :--------: | :--------: | :--------: |
> | HAMT| 66 | 61 | 65 | 60 |
> | Li & Bansal (2023)|  **68** | **62** | 65 |60 |
> | HAMT+ | 67 | **62** | **67** | **62** |
>
> - Following suggestions, **we conduct additional experiments on R2R and REVERIE with DUET as the base model.** We list the model performance on the val-unseen set below. We can see navigation performance improvements of MPM over the strong baseline on these two tasks, demonstrating the effectiveness of MPM. We will include the results in the revised version. Note that currently we only ask the model to reconstruct the masked path and do not ask the model to output object regions, thus the model grounding performance is not improved and we leave incorporating object supervision into MPM as a future direction.
>
> | **Model** | **TL** |  **NE** |  **SR** | **SPL**  |
> | :--------: | :--------: | :--------: | :--------: | :--------: |
> | DUET | 13.94 | 3.31 | 72 | 60 |
> | DUET+MPM | 13.61 | **2.90** | **72.80** | **63.64** |
>
> | **Model** | **TL** |  **OSR** |  **SR** | **SPL**  | **RGS** |  **RGSPL** |
> | :--------: | :--------: | :--------: | :--------: | :--------: | :--------: | :--------: |
> | DUET | 22.11 | 51.07  | 46.98 | 33.73 | **32.15** | **23.03** |
> | DUET+MPM | 22.54 |  **51.41** | **47.57** | **33.91** | 31.87 | 22.67 |
>
>
> We hope the additional experiments and explanations can address your concerns and we are happy to conduct more experiments per your suggestions.

---

### Official Review · Reviewer_Sdnd · 2023-08-07

**Soundness:** 3

**Excitement:**

3: Ambivalent: It has merits (e.g., it reports state-of-the-art results, the idea is nice), but there are key weaknesses (e.g., it describes incremental work), and it can significantly benefit from another round of revision. However, I won't object to accepting it if my co-reviewers champion it.

**Missing References:**

[ECCV'20] Counterfactual Vision-and-Language Navigation via Adversarial Path Sampler, which also considers data augmentation for VLN.



**Paper Topic And Main Contributions:**

This paper introduces masked path modeling (MPM) for vision-and-language navigation (VLN) to address the data scarcity limitation. They first explore and collect navigation paths in the downstream environments. MPM then learns to reconstruct the executed actions of the masked subpath. Experimental results show that MPM can effectively improve the navigator performance across all R2R, R4R, and RxR datasets.


**Questions For The Authors:**

Please see Reasons to Reject

**Reasons To Accept:**

+ This paper is well-written and easy to follow.
+ This MPM framework is simple yet effective and seems to be model agnostic with better scalability.

**Reasons To Reject:**

+ The idea of mask-then-reconstruct for VLN has first been proposed by (Hao et al., 2020), where the difference is to predict missing language tokens or actions of the navigation. I am wondering if this achieves the EMNLP novelty bar.
+ For MPM learning, it only considers the visual observation of the navigation path. How to improve the crucial cross-modal perception in this process? Does MPM just understand the action transformation but not the vision-and-language interaction?
+ In L584, they mention that previous methods are limited by the size of VLN data. However, MPM also only collects randomly-sampled paths in the downstream environment. How does MPM extend to a large-scale setting as they claim?

**Reproducibility:**

4: Could mostly reproduce the results, but there may be some variation because of sample variance or minor variations in their interpretation of the protocol or method.

**Reviewer Confidence:**

4: Quite sure. I tried to check the important points carefully. It's unlikely, though conceivable, that I missed something that should affect my ratings.

---

> ### Author Rebuttal · Authors · 2023-08-29
>
> Thank you for your insightful feedback! We are glad that you find our paper well-written and our framework is simple and effective. Below we address your concerns:
>
> >Difference between Hao et al. (2020) and our work
>
> Thank you for mentioning the relevant work! We would like to emphasize that while one of their objectives is to predict missing actions, their agent is **conditioned on both the instruction and paths** while our MPM objective only requires the model to be conditioned on **self-collected paths.** This design leads to significant differences: their model can only be trained on a limited number of parallel instruction-path pairs, whereas our MPM allows the model to utilize large-scale self-collected data. We have this discussion in the introduction (line 74-80) and related work sections (line 577-586) and we will expand the paragraphs.
>
> > Cross-modal perception in MPM
>
> We agree that cross-modal perception is crucial in navigation. In MPM, the main objective is to connect the vision representations to the output actions with large-scale self-collected data and it is not focused on improving the vision-language interactions. **We believe that improving cross-modal interactions is orthogonal to our work and we leave it as our future work.**
>
> That being said, our vision encoder is initialized with the CLIP model, which is trained with aligned image-text pairs, ensuring a good match between the model vision and language representations. **In addition, we did try adding a contrastive loss between the vision and language encoders.** Specifically, we use the CLIP-style contrastive losses to explicitly align the vision representations of the collected paths and the language representations of the input instructions. However, we did not see improvements with such an objective as shown below. We will include these results in the revised version as a reference for future work.
>
>
> | **Model** | **TL** |  **NE** |  **SR** | **SPL**  |
> | :--------: | :--------: | :--------: | :--------: | :--------: |
> | HAMT+MPM | 11.99 | 3.44  | 68.37 | 62.59 |
> | HAMT+MPM+CLIP | 11.59  | 3.35 | 67.18 | 61.97 |
>
>
>
> > Extending to large-scale settings
>
> We would first like to clarify our claim that previous methods are limited by the size of VLN data. Most of the previous works have to use human-annotated or machine-generated **instruction-path pairs**. Human-annotated data is hard to acquire and is therefore limited in quantity; machine-generated data can sometimes be of low quality. On the other hand, our method only requires the model to predict a self-collected path from the environment given some of its viewpoints masked. Therefore, the agent can collect **a large number of paths from the environment without using an external model** to generate data of questionable quality. We will explain this in detail in the revised version. **In 3.3, we see performance gains when the agent is allowed to explore unseen environments, demonstrating the potential of MPM given more downstream environments without the need to have human-annotated instruction-path pairs.**
>
> We hope the additional experiments and explanations can address your concerns and we are happy to conduct more experiments per your suggestions.

---

### Official Review · Reviewer_rViw · 2023-08-13

**Soundness:** 3

**Excitement:**

3: Ambivalent: It has merits (e.g., it reports state-of-the-art results, the idea is nice), but there are key weaknesses (e.g., it describes incremental work), and it can significantly benefit from another round of revision. However, I won't object to accepting it if my co-reviewers champion it.

**Paper Topic And Main Contributions:**

This paper presents a new objective for VLN training: Mask Path Modeling (MPM), which is to predict the complete path given a randomly masked subpath, to address the data-scarcity problem for training VLN agents. MPM is applied to a history-aware multimodal transformer model in both pre-training and fine-tuning and gets a large performance gain, with +1.3%, +1.1%, and +1.2% SR on the val-unseen split of the R2R, R4R, and RxR datasets, respectively.

**Questions For The Authors:**

My questions are listed in the weakness of the paper.

**Reasons To Accept:**

1. This paper is strongly motivated and well-written.
2. The main idea is to directly use masked paths instead of instructions to guide agent navigation as paths are easier to get than instructions. This idea is interesting because path is a very different modality from natural language instructions.
3. They achieve strong results with MPM on multiple VLN datasets and the authors provide some interesting analysis of the method.

**Reasons To Reject:**

1. MPM is a very general objective for training navigation agents but the author only applies MPM to fine-grained instruction-following tasks. I'd like to see more tasks applying MPM like REVERIE.
2. The improvements reported in the paper are mainly related to the data/training and not to the model, so I expected more "X+MPM" comparison experiments. Like DUET+MPM, EnvDrop+MPM to support the claim in the paper.
3. Some words/claims in the paper are confusing for me. E.g. "Path dynamics" mentioned in the abstract isn't explained well in the main paper (even occurs). The widely-used VLN augmentation data Prevalent also collects "diverse and substantial amount of data/paths" as it samples all the possible paths from MP3D environments.
4. Some experiment results are confusing. E.g. In Figure 3, 100% mask improves ~1% SR (from ~67% to ~68%). But in this case, the agent got no signals from the masked paths, which should introduce noise to the training process. I don't know why it can still improve agent's performance.

While the paper presents some positive results of MPM compared to the baseline, I think the reason behind the observed improvements appears to need some exploration.

**Reproducibility:**

5: Could easily reproduce the results.

**Reviewer Confidence:**

4: Quite sure. I tried to check the important points carefully. It's unlikely, though conceivable, that I missed something that should affect my ratings.

---

> ### Author Rebuttal · Authors · 2023-08-29
>
> Thank you for your detailed feedback! We are encouraged to know that you find our paper strongly motivated and our idea interesting. Below we address your concerns:
>
> > More tasks applying MPM like REVERIE and more “X+MPM” comparison experiments like “DUET+MPM”.
>
> We agree that MPM is a general paradigm and following your suggestions, we perform additional experiments on the REVERIE and R2R tasks with DUET as baseline with minimal parameter tuning. We list the model performance on the val-unseen set below. We can see that MPM can achieve good navigation improvements over the strong baseline on these two tasks, demonstrating the effectiveness of MPM. We will include the results in the revised version. Note that currently we only ask the model to reconstruct the masked path and do not ask the model to output object regions, thus the model grounding performance is not improved and we leave incorporating object supervision into MPM as a future direction.
> | **Model** | **TL** |  **NE** |  **SR** | **SPL**  |
> | :--------: | :--------: | :--------: | :--------: | :--------: |
> | DUET | 13.94 | 3.31 | 72 | 60 |
> | DUET+MPM | 13.61 | **2.90** | **72.80** | **63.64** |
>
> | **Model** | **TL** |  **OSR** |  **SR** | **SPL**  | **RGS** |  **RGSPL** |
> | :--------: | :--------: | :--------: | :--------: | :--------: | :--------: | :--------: |
> | DUET | 22.11 | 51.07  | 46.98 | 33.73 | **32.15** | **23.03** |
> | DUET+MPM | 22.54 |  **51.41** | **47.57** | **33.91** | 31.87 | 22.67 |
>
> >Confusing words/claims
>
> Thank you for your suggestions! We will revise these words and claims carefully in the revised version. In the paper, “path dynamics” mainly refers to the ability to map a given path to a sequence of actions, which is different from previous works such as PREVALENT that generate pseudo instruction-pair data for data augmentation.
>
> >100\% masking still improves the model performance
>
> Thank you for pointing this out! To clarify, we always keep the last viewpoint of the collected path in all the settings. Therefore, the agent is still conditioned on the goal viewpoint in the 100\% masking setting. We will make this clear in the revised version.
>
>
> We hope the additional experiments and explanations can address your concerns and we are happy to conduct more experiments per your suggestions.

---

### Meta-Review · Area_Chair_EVa1 · 2023-09-11

**Recommendation:** 4

**Metareview:**

The paper proposed a new auxiliary task for vision-and-language navigation, and achieves effective improvement over the baseline. I especially appreciate the honesty in both the paper (e.g., using a stronger baseline with latest CLIP model) and the discussions (e.g., acknowledge that 100% masking = img-goal navigation). Overall, most of the reviewer's concern are resolved during the discussion.

P.s., I encourage the authors to incorporate the proposed changes in their final summary, and also resolve the confusion raised by reviewers. Some confusions come from writing (not from results). For these, I did not see a plan of changes in the final summary but I think that these clarifications are important. A paper is born for readers to learn, thus clear writing is also important as the strong results.

---

### Decision · Program_Chairs · 2023-10-07

**Decision:**

Accept-Findings

**Comment:**

The paper proposed a new auxiliary task for vision-and-language navigation, and achieves effective improvement over the baseline. I especially appreciate the honesty in both the paper (e.g., using a stronger baseline with latest CLIP model) and the discussions (e.g., acknowledge that 100% masking = img-goal navigation). Overall, most of the reviewer's concern are resolved during the discussion.

P.s., I encourage the authors to incorporate the proposed changes in their final summary, and also resolve the confusion raised by reviewers. Some confusions come from writing (not from results). For these, I did not see a plan of changes in the final summary but I think that these clarifications are important. A paper is born for readers to learn, thus clear writing is also important as the strong results.